# Second Victims among Austrian Pediatricians (SeViD-A1 Study)

**DOI:** 10.3390/healthcare11182501

**Published:** 2023-09-08

**Authors:** Eva Potura, Victoria Klemm, Hannah Roesner, Barbara Sitter, Herbert Huscsava, Milena Trifunovic-Koenig, Peter Voitl, Reinhard Strametz

**Affiliations:** 1The Second Victim Association Austria, 11900 Vienna, Austria; eva.potura@secondvictim.at (E.P.);; 2Wiesbaden Institute for Healthcare Economics and Patient Safety (WiHelP), Wiesbaden Business School, RheinMain University of Applied Sciences, 65183 Wiesbaden, Germany; 3Training Center for Emergency Medicine (NOTIS e.V), 78234 Engen, Germany; 4First Vienna Pediatric Medical Center, Sigmund Freud University Vienna, 1220 Vienna, Austria

**Keywords:** second victim, emotional distress, pediatricians, pediatrics, emotional burden, psychological distress, patient safety, health care worker safety

## Abstract

(1) Background: The second victim phenomenon (SVP) plays a critical role in workplace and patient safety. So far, there are limited epidemiological data on the SVP in German-speaking countries. Some studies have been carried out in Germany, but so far, no quantitative studies have been carried out in Austria examining the prevalence, symptom load and preferred support measures for second victims (SVs). This study therefore examines the SVP among Austrian pediatricians. (2) Methods: A nationwide, cross-sectional and anonymous online study was conducted using the SeViD questionnaire (Second Victims in Deutschland) including the Big Five Inventory-10 (BFI-10). Statistical analysis included binary-logistic and multiple linear regression with the bootstrapping, bias-corrected and accelerated (BCa) method based on 1000 bootstrap samples. (3) Results: Of 414 Austrian pediatricians, 89% self-identified as SVs. The main cause of becoming an SV was the unexpected death or suicide of a patient. High neuroticism and extraversion values as well as working in outpatient care positively correlated with having experienced the SVP. A preferred support strategy was access to legal counseling. (4) Conclusions: Austrian pediatricians have the highest SVP prevalence measured with the SeViD questionnaire. Further research should focus on prevention strategies and intervention programs.

## 1. Introduction

Healthcare carries significant risks, impacting not only patients but also healthcare professionals [1]. In addition to common risks to physical integrity such as needlestick injuries [2] or psychological stress [1,3,4], unanticipated clinical events and outcomes resulting from healthcare errors not only harm patients but also cause distress and trauma to healthcare professionals, leading to them being referred to as second victims (SVs). The second victim phenomenon (SVP) was first described by Albert Wu in 2000 [5]. In this description, the SVP mainly referred to physicians who after having committed a medical error are themselves traumatized and are therefore the SVs of this error, the patients and their relatives being the first victims. This definition has since been updated and broadened. Scott et al. theoretically conceptualized it further and defined SVs as healthcare workers who are traumatized by an unanticipated adverse event [6]. In 2022 the European Researcher Network Working on Second Victims (ERNST), an EU-COST action, published an evidence- and consensus-based definition of the term: an SV is “any health care worker, directly or indirectly involved in an unanticipated adverse patient event, unintentional healthcare error, or patient injury and who becomes victimized in the sense that they are also negatively impacted [7]”. The psychological reactions as a consequence of the SV experience could range from feelings of guilt, anxiety and loss of confidence to loss of trust in the healthcare system, absenteeism, turnover intentions, alcoholism and even committing suicide [8,9,10]. The SVP not only affects the SVs themselves negatively, but can also compromise the quality of care future patients receive [11], for example, due to the practice of defensive medicine [12,13] or due to heightened error rates after the development of post-traumatic stress disorder (PTSD) [14,15]. The prevalence of the SVP has been researched previously in various studies [16]. So far, three quantitative studies called SeViD-I [17], SeViD-II [18] and SeViD-III [19] about SVP have been conducted and published in Germany, and one qualitative study has been conducted in Western Austria among intensive care nurses [9]. However, no nationwide studies have been conducted to further investigate the SVP in Austria.

In addition, numerous studies sought to determine different risk and protective factors of job-related strain in the past several decades. The identified factors can be roughly divided into (1) demographic; (2) job, workplace, or organizational; and (3) personality trait characteristics. The growing body of evidence suggests that there is a negative correlation between increasing age and job-related stress levels and burnout among different healthcare professionals [20,21,22]. Being a female was found to be a risk factor for developing symptoms of burnout among surgeons [22]. This is in line with the results of SeViD-I on the SVP amongst young German physicians in internal medicine, which suggested that female sex was significantly correlated with a higher probability of becoming an SV and higher levels of perceived symptom quantity and intensity after the SV experience.

Moreover, job, workplace and organizational characteristics could impact psychological as well as physical well-being among healthcare providers according to previous studies [23,24]. In contrast, no association between the type of workplace (working in the operating theater, intensive care unit or prehospital setting) and likelihood of becoming an SV could be found in SeViD-I and SeViD-III. Aside from studies that highlighted a greater risk of experiencing posttraumatic stress, depression and anxiety for pediatricians compared to the general population [25], there is some evidence that pediatricians working in inpatient hospital wards were more frequently affected by burnout than their colleagues working in the outpatient wards [26]. However, associations between the type of care (inpatient vs. outpatient) and frequencies of critical events that could cause the SVP among different healthcare professionals have not yet been explored. 

Furthermore, previous studies suggested that personality traits could play an important role in health impairment as well as motivational processes in the organizational context [27]. Results of SeViD-II and SeViD-III revealed that higher levels of neuroticism were correlated with a higher likelihood of experiencing the SVP or higher symptom load among healthcare professionals that have already experienced the SVP at least once in their carrier (personality trait variables were not collected in SeViD-I). The studies aimed to investigate if personality traits could explain the likelihood of the SVP and the unique proportion of variance of symptom load after the inclusion of demographic and workplace-related variables in the theoretical model since demographic and workplace-related factors were already included in SeViD-I. Moreover, neuroticism partially mediated the relationship between the length of professional experience in years and symptom load after the SV experience in SeViD-III. In this study, the magnitude of the negative correlation of the length of professional experience to symptom load was reduced after the inclusion of neuroticism in the regression model. The post hoc analysis showed that neuroticism was indeed a mediating variable in the relationship between the length of professional experience and symptom load: longer professional experience was associated with lower levels of neuroticism which were in turn associated with lesser symptom load. The possible explanation for this finding was that the length of professional experience could be considered a potential resource in the context of the job demands–resources model which postulates that high job demands in combination with low job resources could lead to burnout and other consequences of the health impairment process [28]. In addition, the model emphasizes the positive associations between neuroticism and job demands, as well as the negative associations between neuroticism and job resources [27,29]. In this respect, healthcare professionals with longer professional experience might have modified their coping styles towards more adapted coping over the years and built a network of colleagues that could provide informal emotional and social support, as opposed to their colleagues with less professional experience [30]. Hence, we assumed that the length of professional experience could act as a resource and reduce the SV-related symptom load. However, coping styles as well as prosocial behaviors are related to personality characteristics [31]. Therefore, we aimed to a priori test the effect of length of professional experience on symptom load after SV experience via each of the Big Five personality traits in the current study.

To summarize, the study’s primary aim is to assess the prevalence, symptom load and preferred support strategies for SVP and to identify demographic, workplace-related and personality trait factors that significantly correlate with the likelihood of experiencing the SVP and the amount and severity of possible symptoms caused by the SVP amongst Austrian pediatricians. Lastly, our aim is to test the mediational model that postulates that length of professional experience can indirectly reduce symptom load caused by the SV experience through the Big Five personality traits. This study was conducted by the Austrian Second Victim Association and the Wiesbaden Institute for Healthcare Economics and Patient Safety (WiHelP) in cooperation with the Austrian Society for Pediatrics and Adolescent Medicine (ÖGKJ).

## 2. Materials and Methods

### 2.1. Design and Conduction of the SeViD-A-I Survey

This study was a nationwide cross-sectional study conducted among Austrian pediatricians using the SeViD questionnaire (Second Victims in German-speaking Countries). The development and validation of the questionnaire are described in detail in another publication [32]. The questionnaire consisted of 25 questions covering seven dimensions, basic demographics, knowledge and exposure to the SVP, the adverse event leading to it, the recovery process, reactions, support measures and and the 10-item short-version of the Big Five Inventory (BFI-10) to assess the personality traits of the participants (openness, neuroticism, agreeableness, extraversion and conscientiousness) [32]. The questionnaire was slightly modified: two questions were added, firstly if the adverse event leading to the SVP was related to the COVID-19 pandemic and secondly if those affected feared legal consequences. 

The survey was distributed via email among the Austrian Association for Pediatric and Youth Medicine (ÖGKJ), representing approximately 2100 members. The invitation letter included a short overview of the aims of the study, a brief introduction to the SVP and a link as well as a QR code leading to the online survey using the SurveyMonkey platform. The data were collected anonymously; tokens, cookies and IP addresses were not stored. The study period was seven weeks (between 14 March and 1 May 2023); reminders were sent out after six weeks. The survey used adaptive questioning: only those who identified themselves as SVs were given the questions referring to the events leading to the SVP or their symptoms. All questions needed to be answered before a progression to the next item was possible, though the participants could revise their previous answers and change them, if necessary. Also, they had the opportunity to comment on the questionnaire or make suggestions at the end of the survey. 

### 2.2. Preparation and Re-Coding of Variables for Statistical Analysis

For statistical analysis, the items workplace and SV status were dichotomized. Participants working in an inpatient setting (hospital, rehabilitation) were assigned a value of one, and those working in outpatient (single or group practices) care were assigned a value of zero. Participants who had experienced the SVP once or several times were assigned a value of one, and participants who did not were assigned a value of zero. 

To assess symptom load, sum scores of 18 different symptoms that can be directly caused by the SVP were determined according to the previous publication. Our aim was to assess the expression of every symptom that can be a direct consequence of SV and to compute a sum of the expressions of all 18 symptoms. In this way, we wanted to obtain an overall measurement of the symptom load as the focus of the present study was not to find predictors significantly associated with each individual symptom but to find predictors significantly associated with the overall symptom load. Sum scores of symptom load measurements are widely used in the literature since they are easier to handle and calculate compared to mean scores in (clinical) practice [33,34]. Although our study and previous surveys were not conducted in the clinical setting, we elected to use sum instead of mean scores since the measurement might be transferred and used in clinical practice in the future. We were able to statistically calculate sum scores because we used only items with positive scoring measuring symptom load. For symptom load, the participants were assigned the following values according to the previous SeViD studies: “strongly pronounced” was assigned a value of “1”, the option “weakly pronounced” was assigned a value of “0.5”, and “not at all” and “I don’t know” were assigned a value of “0”. The reasoning behind merging “not at all” and “I don’t know” into one category was that we assumed that the expression of the certain symptom among individuals choosing the response option “I don’t know” might not exceed the benchmark of “weakly pronounced” as we expected that participants with at least clearly weakly pronounced symptom expression would be able to choose the other two response alternatives. However, this practice might be disputable as participants choosing this option might be uncertain about their feelings or have difficulty comprehending the displayed symptom. Therefore, we performed analyses twice: merging categories “not at all” and “I don’t know” with the assigned value of 0 in one case and treating the “I don’t know” option as a missing value in the other. We report on the latter group of analyses in the Appendix A. Additionally, three more symptoms were presented to the participants that were not included in the sum score: “the desire to be supported by others”, “the desire to process the event for better understanding” and “fear of legal consequences”. “The desire to be supported by others” and “the desire to process the event for better understanding” were not included in computing the overall sum score, because they were somewhere in between symptoms and support desires according to the SeViD-III study. “Fear of legal consequences” was first introduced as a possible symptom in this study and was not a part of the symptom load sum score in the previous SeViD surveys. However, we report on their frequency distributions in the results section referring to descriptive statistics of the individual symptoms.

Preferred support measures were measured using 13 different items on a five-point descending scale ranging from one, “very helpful”, to five, “not helpful at all” (“not helpful at all” = 5, “rather not helpful” = 4, “neutral” = 3, “rather helpful” = 2 and “very helpful” = 1). The participants were asked to rate the support measures even if they did not experience the SVP; i.e., the items referring to support measures were presented to all participants regardless of their SV status. The participants who did not experience the SVP were given the opportunity to reflect on the usefulness of the support measures in the hypothetical case that they become SVs. In addition, we performed the Mann–Whitney test to compare the differences in the ratings of each support measurement between the groups based on the SV status (SVP experienced at least once, yes vs. no).

Time from critical event to full recovery from the SV experience was recorded as follows: one, “less than a day”; two, “within a week”; three, “within a month”; four, “within a year”; five, “more than a year”; and six, “I did not fully recover till now”. It is notable that the item referring to time from critical event to full recovery was treated as a nominal scaled variable indicating that the assigned values do not necessarily demonstrate an ordered range since the response option “I did not fully recover till now” could have been theoretically shorter than other response options considering the time since the critical event occurred.

### 2.3. Statistical Analysis

In order to test factors associated with the occurrence rate of SV and symptom load caused by SV experience, we performed two regression analyses with the same nine predictors, namely demographic predictors (gender, age, length of professional experience); a dichotomized predictor indicating workplace, i.e., type of care (inpatient vs. outpatient); and five personality traits (openness, neuroticism, agreeableness, extraversion and conscientiousness) by including demographic characteristics, type of workplace and finally personality traits in the regression equations in line with the aims of our study and the two previously published German national surveys on the SVP. No predictor variable served as a control variable. Possible associations between the predictors gender, age, length of professional experience, workplace (inpatient vs. outpatient) and five personality dimensions (openness, neuroticism, agreeableness, extraversion and conscientiousness) with the dichotomous outcome variable SV status (yes vs. no) were assessed using binary logistic regression with the bootstrapping, bias-corrected and accelerated (BCa) method based on 1000 bootstrap samples.

In addition, we used multiple linear regression with bootstrapping, the BCa method, based on 1000 bootstrap samples to test if the predictors gender, age, length of professional experience, workplace (inpatient vs. outpatient) and personality traits (openness, neuroticism, agreeableness, extraversion and conscientiousness) can significantly contribute to the explanation of variance in symptom load among participants who previously identified themselves as SVs. We inspected the bivariate correlation matrix as well as tolerance and the variance-inflation factor (VIF) to test if multicollinearity between the predictors was present. VIF values above 5 were considered problematic, and values above 10 indicated a serious collinearity problem [35,36]. Additionally, a value of 0.1 was considered the minimum level of tolerance [36]. Results of the multicollinearity diagnostic showed that the variables age (VIF = 7.12, tolerance = 0.14) and length of professional experience (VIF = 6.91, tolerance = 0.15) showed considerable collinearity according to VIF, and these variables were mean-centered before the analysis was conducted.

Considering that the post hoc analysis in SeViD-III revealed that the length of work experience had an indirect effect on symptom load via neuroticism, we decided to test the indirect effect of work experience on symptom load via each of the five personality traits, namely openness, neuroticism, agreeableness, extraversion and conscientiousness, in the current study. For this purpose, we used model 4 for parallel mediators of the PROCESS macro for SPSS v4 [37]. The model automatically applies bootstrapping using the deviation-correction method at the 95% confidence intervals based on 5000 bootstrap samples to estimate direct, indirect and total effects. Figure 1 displays the theoretical model. 

Descriptive statistics of demographic variables and applied instruments were reported using means (M) with standard deviations (SD) for interval-scaled and frequencies (*n*) with percentages (%) for nominal and ordinal scaled variables. Statistical analysis was performed using SPSS Statistics Version 29 (IBM, New York, NY, USA). A *p*-value lower than 0.05 was considered significant. 

## 3. Results

### 3.1. Descriptive Baseline Analysis

Of the 2100 participants addressed, 414 responded to the questionnaire (response rate 19.7%). Eighty-five percent of the respondents fully completed the survey. The mean time for participation was 9 min and 27 s. Table 1 shows the participants’ baseline characteristics.

Most participants were female (70%), 30% were male and none of the participants stated having a diverse (non-binary) gender identity. The mean age of the participants was 45.38 years (SD 11.03; range 23–67). The mean work experience was 15.62 years (SD 10.60; range 1–47). 

When asked to indicate the type of their workplace, 166 (40.09%) participants stated that they worked in outpatient care in either single or group practices. All others reported working in a hospital or rehabilitation setting. Most (71.98%) reported working full-time.

Concerning their knowledge of the term “Second Victim”, 73.73% (306) of the participants indicated not having heard of it prior to the survey. After the term was explained briefly, 89% (365) reported that they would identify themselves as SVs, as shown in Figure 2. Most have experienced being a second victim more than once (74%). Eleven percent of participants reported not having experienced the SVP, and 59.94% stated that they had become SVs within the past twelve months.

Regarding the events leading to SV traumatization, most participants stated that the unexpected death or suicide of a patient was their key event (33.06%), followed by the aggressive behavior of patients or relatives (23.06%), as shown in Table 2. Of all events, 7% of the key incidents leading to the SVP were related to the COVID-19 pandemic. 

Some specified the key events further:


*“Death of a child reminding me of my daughter”*

*“Severely abused child”*

*“Case for child protective services”*

*“Uncalled-for legal action from the patient which affected me personally”*

*“Ending therapeutic measures in a palliative setting (ventilation)”*


Over half of the participants (54.17%, 195) received help after those traumatizing adverse events; 38.33% (138) did not receive any help but had not asked for it, while 7.5% (27) stated they were denied help after actively asking for it. The vast majority received help from their colleagues, as shown in Table 2.

More than half of the participants recovered from the SVP within a month (66.16%); 5.49% reported needing more than a year to recover, whereas 14.02% (18 in total) reported having not yet fully recovered. 

Regarding their reactions to the adverse event, the participants were given 21 possible symptoms (see Table 3 for 18 symptoms that were included in the symptom load sum score) and were asked to rate them based on how pronounced they felt these symptoms were for them. 

The most pronounced symptoms were therefore self-doubts (30.4%), flashbacks within similar professional situations (26.1%) and insomnia or excessive need for sleep (25.6%). Additionally, when the participants were asked about the desire to be supported by others, 32.5% indicated that this desire was strongly pronounced. Also, for 38.3%, the desire to process the event for better understanding was strongly pronounced. When the participants were asked about the fear of legal consequences, the majority (65.8%) indicated, that this was weakly or not at all pronounced. Only 12.3% indicated that the fear of legal consequences was strongly pronounced.

Table 4 shows how the participants rated possible support measures (1 = very helpful, 5 = not helpful at all).

The two groups of participants significantly differed in rating 5 of 13 support measures. Participants who reported they have not experienced the SVP rated the measures of the possibility of taking time off from work directly to process the event, access to professional counseling or psychological/psychiatric consultations (crisis intervention), and clear and timely information regarding the course of action after a serious event (e.g., damage analysis, error report) to be more helpful after the adverse event. In contrast, participants who identified themselves as SVs rated support (mentoring) when continuing to work with patients and guidelines regarding the role (activities) expected of them during a serious event as more helpful when compared to the participants who have not experienced the SVP.

### 3.2. Factors Associated with the Likelihood of Becoming a Second Victim

The results of binary logistic regression showed that only two of five personality traits, extraversion and neuroticism, were significant risk factors for experiencing the SV effect (extraversion: regression coefficient B = 0.44, BCa 95% CI [0.04, 0.95], odds ratio (OR) 1.55, 95% CI [1.06, 2.28]; neuroticism: B = 0.55, BCa 95% CI [0.07–1.20], OR = 1.73, 95% CI [1.13, 2.64]). The results of binary logistic regression of potential risk and protective factors for becoming an SV are shown in Table 5.

### 3.3. Correlation Matrix

Prior to conducting the multiple linear regression analysis, we inspected the bivariate correlations between all variables included in the analysis. Table 6 displays the Pearson’s product-moment correlation matrix with BCa 95% CIs based on 1000 bootstrap samples. 

Besides the strong correlation between age and the length of professional experience (r = 0.92, *p* < 0.01), all other correlations were weak. Symptom load, the variable that served as a criterion in multiple linear and mediational analyses, was significantly negatively correlated with the length of professional experience (r = −0.13, *p* < 0.05) and extraversion (r = −0.15, *p* < 0.01) and significantly positively correlated with neuroticism (r = 0.25, *p* = 0.01).

### 3.4. Factors Associated with the Symptom Load after the SV Experience

Results of multiple linear regression showed that besides the personality traits openness, extraversion and neuroticism, the type of workplace had a significant contribution to the explanation of variance in the symptom load after the SV experience in 313 physicians who completed the survey and reported encountering the SVP at least once in the career. The response options “Not at all” and “I don’t know” were treated as the same category in the computation of the symptom load. Pediatricians working in outpatient care had a higher symptom load compared to those who worked in an inpatient setting. Neuroticism and openness were significantly correlated to a higher symptom load. In contrast, pediatricians with higher levels of extraversion had a lower symptom load (see Table 7).

In addition, similar results were shown in the multiple linear regression with symptom load as the criterion variable that was computed using the following scoring of participants’ answers: “Not at all” = 0, “weakly pronounced” = 0.5 and “strongly pronounced” = 1, while the response option “I don’t know” was considered a missing value (see Appendix A).

### 3.5. Testing the Mediational Model

We performed the mediation analysis with professional experience as a predictor, the Big Five dimensions as mediators and symptom load as an outcome. As shown in the correlation matrix, length of professional experience correlates significantly with symptom load (see Table 6). 

The results revealed a negative significant indirect effect of professional experience on symptom load (unstandardized regression coefficient (B) = −0.01, bootstrapped 95% CI [−0.02, −0.003]; standardized regression coefficient (β) = −0.04, bootstrapped 95% CI [−0.07, −0.01]) via neuroticism. Furthermore, the direct effect of professional experience on symptom load in the presence of the mediator neuroticism was found to be non-significant (B = −0.03, bootstrapped 95% CI [−0.06, 0.001]; *p* = 0.01). Hence, neuroticism totally mediated the relationship between professional experience and symptom load caused by the SV experience. The length of professional experience showed no further significant indirect effects on symptom load via the other four personality traits (see Table 8 and Table 9). The total (indirect + direct) effect of length of professional experience on symptom load was B = −0.04.

## 4. Discussion

Our study aimed to investigate the frequency and severity of the SVP among Austrian pediatricians. The survey revealed that 89% of Austrian pediatricians who completed the survey reported that they have experienced the SVP at least once in their professional careers. This exceeds the results of international studies [16] as well as the results of our three previous SeViD studies using the same questionnaire, which showed prevalences of 60%, 59% and 53%, respectively, therefore being the highest sample proportion of SVs measured yet in German-speaking countries. A high prevalence was to be expected from previous SeViD studies, though not as high as this study’s 89%. This might refer to a high proportion of pediatricians working in outpatient settings participating in this study. A study by Wolf et al. conducted among pediatric intensive care providers also deduced that pediatric healthcare workers were at higher risk of experiencing the SVP [38]. Another study showed that pediatric healthcare workers experienced significantly more symptoms of secondary traumatic stress and had lower resilience than population means [39] possibly explaining the high prevalence of the SVP among this study’s pediatric physicians. Another possible explanation is the increasing awareness of the impact of medical errors on healthcare professionals within the past ten years. In this study, we showed that although the majority of pediatricians were not familiar with the term SVP, they were generally able to understand the presented definition of the construct, reflect on their employment history and draw conclusions about the possible critical event and their psychological reactions to this experience. This indicates that this phenomenon is present among the target population even though approximately three-quarters of them were not aware of the meaning of the term. Nevertheless, we believe that the trend towards the broader usage of the term in the practice could become imminent as the awareness of the phenomena in the population of healthcare professionals increases, which, for example, is the case in the research [40]. 

The most prominent key event leading to the SVP was the unexpected death/suicide of a patient (33%), followed by aggressive behavior of patients/relatives (23%) and near misses (19%). This not only correlates with our previous SeViD studies, but also emphasizes the significance of near misses as a potential event causing the SVP. The added question of whether the key event leading to the SVP was related to the COVID-19 pandemic showed that the pandemic did not trigger the majority of the SVPs. Of course, the pandemic had a negative impact on healthcare workers’ mental health, leading to post-traumatic stress disorder or depression [41]. But in the case of the SVP, it seems to only play a minor role. The previous SeViD studies have come to the same result: SeViD-I was conducted before the pandemic, and the prevalence in SeViD-I is similar to that in SeViD-II and -III, which were conducted during the pandemic. Therefore, the pandemic might be seen as an accelerator but not the main trigger of the SVP.

The by far most important group for the participants of this study who identify as SVs when seeking out help is their colleagues (87.71%). This not only aligns with previous studies but also, in combination with the self-perceived time of recovery being mainly low (66% recovered within a month), aligns with the Scott Three-Tiered Interventional Model of Support which proposes peer support as a very effective measure when dealing with the SVP and being sufficient for 60% of SVs [42]. 

The rating of possible support measures showed that the most helpful support measure is the possibility of accessing legal consultation after a severe event. This contrasts with 65% of the participants stating that fear of legal consequences was only weakly if at all pronounced for them after experiencing the SVP. Perhaps, in general, the fear of legal consequences is high but did not apply to the situations that caused the participants to become SVs. However, there was a significant difference between participants who had experienced the SVP and those who had not. Participants who had not experienced the SVP tended to rate this type of support higher compared to the participants who had experienced the SVP at least once. Still, this should be kept in mind when preparing a support program for Austrian pediatricians, and access to legal advice should be offered after a severe adverse event complementary to systematic peer support, as suggested in Scott’s model [42]. Also, there is the possibility to discuss emotional thoughts and be provided with clear and timely information regarding the course of action after a serious event. This can be used to tailor suitable intervention programs for SVs in Austria, though further studies are necessary to confirm this since the results of this study cannot be generalized. However, participants who identified themselves as SVs reported that concrete workplace-related support measures such as support (mentoring) when continuing to work with patients and guidelines regarding the role (activities) expected of them during a serious event would be more appropriate than general measures like psychological or legal counseling.

The examination of the Big Five personality traits in regard to the SVP showed that neuroticism positively correlated with the likelihood of becoming an SV, which was also discovered in SeViD-II and -III: participants with higher neuroticism levels were more likely to be an SV than those with lower levels. A new finding of this study is that extraversion, though shown to positively correlate with becoming an SV, seems to negatively correlate with the symptom load when an individual is affected by the SVP. As it is generally agreed that extraverted individuals are resilient and might cope with stress in an effective manner, the research results examining the relationship between extraversion and perceived stress and its consequences are not that conclusive. On the one hand, the findings suggest the positive effect of extraversion on psychological well-being and self-esteem and the negative effect on depression [43]. On the other hand, there is some evidence that personality traits do not play a decisive role in stress perception [44]. Furthermore, extraverted individuals experienced more stress during the COVID-19 pandemic as they were more affected by social distancing, considering that sociability is one of the main facets of this personality dimension [45]. Our results now provide evidence that the role of extraversion might be far more complex in the context of the SVP. One possible explanation for the controversial results regarding extraversion might be that extraverted physicians were more open to acknowledging that they did experience this kind of traumatization. However, it seems reasonable that they could mobilize their resources and ensure social support to genuinely process the critical event and to experience lower levels of symptom load. A similar conclusion was reached by Calvete et al.: Extraverted individuals tended to identify themselves more frequently as victims of bullying. In contrast, extraverted individuals presented a greater reduction in depressive symptoms as a consequence of victimization [46].

Neuroticism and openness seemed to positively correlate with symptom load. A similar pattern of results regarding the correlation between openness, neuroticism and symptom load was obtained in the SeViD-II study amongst German nurses. Here, resilience might play an important role in explaining resistance and therefore a lower symptom load [47]. Resilience is found to negatively correlate with neuroticism and positively correlate with the other personality traits [48,49]. This explains the higher symptom load for participants reaching higher neuroticism scores and lower symptom loads for extraverted participants. It does not, however, explain the higher symptom load for participants with the personality trait openness, though according to Oshio et al.’s meta-analysis, openness showed a weaker positive correlation with resilience than extraversion (r = 0.34 for openness; r = 0.42 for extraversion) [48]. A possible explanation might be that those with greater openness might be more likely to explore alternative treatments or interventions. If these interventions or treatments are not as established, they might bear higher risks to patients and therefore lead to more adverse events when using them. This could not only trigger SVP but also increase the symptom load since those with higher levels of openness might feel more guilty after choosing the riskier treatment or intervention for their patients. Of course, the finding of openness leading to higher symptom load might also be based on coincidence and should be further examined in future studies. These findings of personality traits playing an important role not only in the development of the SVP but also in the symptom load stress the importance of tailored intervention programs for healthcare providers that utilize different prevention, screening and coping strategies.

Another finding is that higher levels of neuroticism were associated with both a shorter length of professional experience and a higher level of symptom load; i.e., length of professional experience and symptom load shared a common proportion of variance with neuroticism, as shown in Table 8 and Table 9. One possible explanation is that, as mentioned above, neuroticism negatively affects resilience [48,49]. Therefore, it is possible that those scoring higher in neuroticism are more likely to leave their jobs, meaning that a selection effect might have taken place for those with higher work experience. This would indicate that those with higher work experience score lower on neuroticism than those with lower work experience. A study conducted among Austrian nurses has shown that 42.4% think about quitting their jobs regularly [50]. Perhaps those scoring higher in neuroticism have quit, hence the symptom load being lower with higher work experience. This selection effect might be the reason for the mediation. Another possibility is the response shift of those with higher work experience: they might have witnessed SVP in other colleagues which led them to put their own experience in perspective, explaining the lower symptom load. Nevertheless, due to the correlational nature of the study, we cannot be sure if the assumed causal direction is correct. Theoretically speaking, physicians who have made medical mistakes can subsequently experience higher levels of anxiety and regret leading to higher symptom load. Symptom load might be in turn expressed in higher neuroticism scores. However, research in the area of personality psychology indicates that personality traits remain relatively stable over the lifespan among the adult population [51,52]. Moreover, neuroticism was found to remain relatively stable even after a traumatic event [53]. In conclusion, we believe that our interpretation of neuroticism affecting the likelihood of SV experience and the symptom load is theory-driven and therefore more plausible than vice versa. Furthermore, length of professional experience might act as a certain resource, and it might be positively connected to adaptive coping style, resilience, social support and other constructs that might help individuals to recover from a traumatic experience.

Multiple linear regression also showed that the workplace had a significant influence on symptom load and that pediatricians in outpatient care had a significantly higher symptom load than those working in a hospital or rehabilitation setting. This might be due to them working mainly in single practices and not being able to debrief with colleagues as easily as those working in a hospital or rehabilitation setting. Since peer support plays an important role in dealing with the SVP [42], not having access to it may be the reason for the significantly increased symptom load for outpatient pediatricians. This indicates the urgent need for systematic, nationwide external support for healthcare workers in Austria. 

The study is not without limitations. Of the approximately 2100 contacted members of ÖGKJ, only 414 participated in this study. Fosnacht et al. showed that the response rate of 20% could provide reliable estimates in medium sample sizes (*n* = 300–500) with the application of the convenience sampling method [54]. Though this is a reasonable response rate for this voluntary study, the possibility of convenience sample bias exists. Therefore, an overestimation of the prevalence of the SVP is possible. Also, the cross-sectional design of the study only enables us to describe associations, but not causations. Additionally, by using adaptive questioning, the study does not allow comparison between those participants who have experienced the SVP and those who have not. Further limitations are described in the SeViD-I and -II studies. More studies with larger sample sizes should be carried out to confirm this study’s findings [55].

## 5. Conclusions

The high prevalence of the SVP in combination with the severity of symptom load among Austrian pediatricians found in this study makes it necessary to acknowledge the SVP as a structural and frequently occurring problem. Therefore, appropriate measures including primary, secondary and tertiary prevention need to be established not only to ensure healthcare workers’ well-being but also to ensure individual and systemic medical care security.

Since outpatient care pediatricians seem to be more severely affected by the SVP, systemwide external support measures and peer support programs should be established. For inpatient care pediatricians, systematic peer support within their organizations should be implemented.

## Figures and Tables

**Figure 1 healthcare-11-02501-f001:**
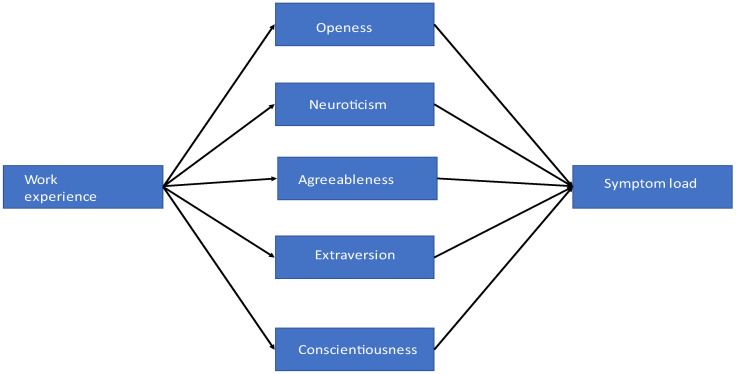
Parallel-mediation model. Work experience: length of professional experience in years; openness, neuroticism, agreeableness, extraversion and conscientiousness: Big Five personality traits; symptom load: the sum of symptoms after second victim experience.

**Figure 2 healthcare-11-02501-f002:**
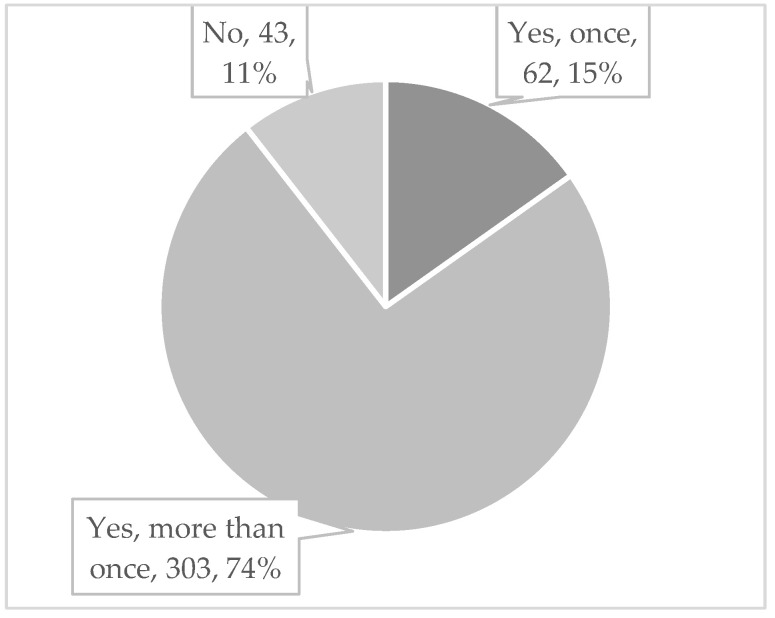
Have you ever experienced a second victim traumatization yourself?

**Table 1 healthcare-11-02501-t001:** Baseline characteristics of the participants.

Characteristics		Percentage of Participants (*n*)
Gender	Female	70 (290)
	Male	30 (124)
Age	20–30 years	10.6 (44)
	31–40 years	25.1 (104)
	41–50 years	31.0 (128)
	51–60 years	22.7 (94)
	>60 years	10.6 (44)
Work experience	1–5 years	20.8 (86)
	6–10 years	17.4 (72)
	11–15 years	18.6 (77)
	>15 years	43.2 (179)
Specialization	None	45.65 (189)
	Neonatology/Pediatric ICU	17.15 (71)
	Pediatric Neurology	9.42 (39)
	Other	27.78 (115)

**Table 2 healthcare-11-02501-t002:** Types of the most formative adverse event (key experience) and most important support groups for SVs.

Type of Event	Percentage of Participants (*n*)
Incident with patient harm	15.83 (57)
Incident without patient harm (near miss)	18.61 (67)
Unexpected death/suicide of a patient	33.06 (119)
Unexpected death/suicide of a colleague	3.89 (14)
Aggressive behavior of patients or relatives	23.06 (83)
Other	5.56 (20)
(Occupational) Group	Percentage of Participants (*n*)
Colleagues	87.71 (157)
Supervisors	25.70 (46)
Management	4.47 (8)
Family/Friends	49.72 (89)
Counselors/Psychotherapists/Psychologic Counseling	19.55 (35)

**Table 3 healthcare-11-02501-t003:** Frequencies of severeness of different symptoms experienced after SV event (*n* = 328).

Symptom	I Don’t Know	Not at All	Weakly Pronounced	Strongly Pronounced
Fear of exclusion by colleagues	15 (4.6%)	215 (65.5%)	78 (23.8%)	20 (6.1%)
Fear of losing the job	8 (2.4%)	245 (74.7%)	60 (18.3%)	15 (4.6%)
Listlessness	10 (3%)	157 (47.9%)	117 (35.7%)	44 (10.6%)
Depressive mood	3 (0.9%)	101 (30.8%)	173 (52.7%)	51 (15.5%)
Concentration difficulties	5 (1.2%)	153 (46.6%)	126 (38.4%)	44 (13.4%)
Reliving the situation outside of professional life	10 (3%)	123 (37.1%)	113 (34.5%)	82 (25%)
Reliving the situation in similar professional situations	12 (3.7%)	69 (21%)	139 (42.4%)	108 (32.9%)
Aggressive, risky behavior	10 (2.4%)	289 (88.1%)	22 (6.7%)	7 (2.1%)
Defensive, overly cautious behavior	12 (3.7%)	90 (27.4%)	152 (46.3%)	74 (22.6%)
Psychosomatic reactions (head- or backaches)	26 (7.9%)	129 (39.3%)	115 (35.1%)	58 (17.7%)
Insomnia or excessive need for sleep	9 (2.7%)	81 (24.7%)	132 (40.2%)	106 (32.3%)
Use of alcohol/drugs because of event	4 (1.2%)	271 (82.6%)	47 (14.3%)	6 (1.8%)
Feelings of shame	11 (3.4%)	183 (55.8%)	93 (28.4%)	41 (12.5%)
Feelings of guilt	2 (0.6%)	111 (33.8%)	125 (38.1%)	90 (27.4%)
Self-doubts	1 (0.3%)	64 (19.5%)	137 (41.8%)	126 (38.4%)
Social isolation	9 (2.7%)	241 (73.5%)	64 (19.5%)	14 (3.4%)
Anger towards others	7 (2.1%)	158 (48.2%)	113 (34.5%)	50 (15.2%)
Anger towards myself	9 (2.7%)	178 (54.3%)	97 (29.6%)	44 (13.4%)

Note. Only symptoms included in the symptom load sum score are listed in the table.

**Table 4 healthcare-11-02501-t004:** Differences in rating of possible support measures after an adverse event in the group of participants that reported that they have already experienced the SVP and the group of participants that reported that they have not experienced the SVP in their career.

Support Measure	M in the Whole Sample	SD in the Whole Sample	M_1_ among SVs	SD_1_ among SVs	M_2_ among Non-SVs	SD_2_ among Non-SVs	*p*
The possibility to take time off from work directly to process the event	2.35	1.23	2.39	1.22	1.97	0.20	0.01
Access to professional counseling or psychological/psychiatric consultations (crisis intervention)	1.68	0.96	1.73	0.98	1.28	0.60	0.00
The possibility to discuss my emotional/ethical thoughts	1.52	0.85	1.53	0.87	1.44	0.64	0.86
Clear and timely information regarding the course of action after a serious event (e.g., damage analysis, error report)	1.50	0.91	1.52	0.87	1.12	0.41	0.00
Formal emotional support in the sense of organized collegial help	1.73	0.97	1.73	0.98	1.64	0.87	0.59
Informal emotional support	1.86	0.98	1.89	1.00	1.64	0.81	0.15
Quick processing of the situation/quick crisis intervention (in a team or individually)	1.47	0.85	1.49	0.85	1.33	0.77	0.15
Support/Mentoring when continuing to work with patients	1.91	1.00	1.03	1.00	1.67	0.92	0.06
Support when communicating with patients and/or relatives	1.93	0.99	1.95	1.00	1.74	0.88	0.21
Guidelines regarding the role/activities expected of me during a serious event	2.06	1.13	2.11	1.16	1.56	0.64	0.00
Support to be able to take an active role in the processing of the event	1.82	0.98	1.85	0.99	1.56	0.82	0.05
A secure possibility to give information on how to prevent similar events in the future	1.60	0.92	1.64	0.96	1.31	0.46	0.01
The possibility to access legal consultation after a severe event	1.37	0.74	1.40	0.77	1.15	0.37	0.00

SV: second victim; M and SD: mean and standard deviation of the participants who completed the survey (*n* = 353) regardless of the second victim status; M_1_ and SD_1_: mean and standard deviation of the participants who reported that they have already experienced the second victim phenomenon; M_2_ and SD_2_: mean and standard deviation of the group of participants who reported that they have not experienced the second victim phenomenon in their career; *p*: a *p*-value of the Mann–Whitney U-test comparing the rating of the support measure between the group of participants who reported that they have already experienced the second victim phenomenon and the group of participants that reported that they have not experienced the second victim phenomenon in their career; support measures were scored on a five-point descending Likert scale ranging from one, “very helpful”, to five, “not helpful at all” (“not helpful at all” = 5, “rather not helpful” = 4, “neutral” = 3, “rather helpful” = 2 and “very helpful” = 1).

**Table 5 healthcare-11-02501-t005:** Factors associated with the likelihood of becoming an SV. Results of binary logistic regression.

Predictor	Regression Coefficient B with BCa 95% CI	*p*	Odds Ratio (Exponentiation of the B Coefficient (Exp(B))	Odds Ratio 95% CI Lower	Odds Ratio 95% CI Upper
Gender ^1^	−0.16 [−1.04, 0.86]	0.68	0.85	0.40	1.82
Age	−0.02 [−0.12, 0.08]	0.68	0.98	0.91	1.06
Professional experience (years)	0.05 [−0.05, 0.17]	0.27	1.05	0.97	1.14
Workplace ^2^	−0.30 [−1.07, 0.46]	0.43	0.74	0.35	1.56
Openness	0.03 [−0.34, 0.40]	0.85	1.03	0.72	1.47
Conscientiousness	0.11 [−0.45, 0.59]	0.66	1.11	0.68	1.82
Extraversion	0.44 [0.04, 0.95]	0.01	1.55	1.06	2.28
Agreeableness	−0.08 [−0.62, 0.41]	0.68	0.92	0.60	1.41
Neuroticism	0.55 [0.07, 1.20]	0.01	1.73	1.13	2.64

Outcome is experiencing second victim status (dichotomous yes = 1 vs. no = 0); ^1^ reference category is male; ^2^ reference category is inpatient care; BCa 95% CI: bias-corrected and accelerated bootstrapping 95% confidence intervals based on 1000 bootstrap samples.

**Table 6 healthcare-11-02501-t006:** Person’s product-moment correlation matrix.

Variable	Gender	Age	Professional Experience	Extraversion	Agreeableness	Conscientiousness	Neuroticism	Openness	Symptom Load
Gender	1								
Age	−0.17 **	1.00							
Professional experience	−0.18 **	0.92 **	1.00						
Extraversion	−0.03	−0.02	−0.03	1.00					
Agreeableness	−0.04	−0.05	−0.04	0.06	1.00				
Conscientiousness	−0.13 *	0.08	0.08	0.13 *	0.04	1.00			
Neuroticism	−0.14 *	−0.13 *	−0.17 **	−0.25 **	−0.03	−0.14 *	1.00		
Openness	−0.07	0.12 *	0.11 *	0.14 *	0.05	0.21 **	−0.16 **	1.00	
Symptom load	−0.04	−0.09	−0.13 *	−0.15 **	0.05	−0.01	0.25 **	0.06	1.00

*: Correlation is significant at the 0.01 level; **: correlation is significant at the 0.05 level.

**Table 7 healthcare-11-02501-t007:** Factors contributing to the explanation of variance in symptom load. Results of multiple linear regression, *n* = 313.

Independent Variable	Unstandardized Regression Coefficient B	Standardized Regression Coefficient β	*p*	BCa 95% CI Lower	BCA 95% CI Upper
Constant	2.41		0.24	*−*1.50	6.10
Gender (female = 1, male = 2)	0.15	0.02	0.70	*−*0.73	0.97
Age ^1^	0.06	0.19	0.26	*−*0.05	0.15
Professional experience (years) ^1^	*−*0.07	*−*0.24	0.13	*−*0.16	0.02
Workplace (inpatient = 1, outpatient = 0)	*−*0.86	*−*0.13	0.02	*−*1.57	*−*0.17
Openness	0.35	0.11	0.03	0.03	0.69
Conscientiousness	0.08	0.02	0.73	*−*0.42	0.58
Extraversion	*−*0.40	*−*0.12	0.04	*−*0.81	*−*0.01
Agreeableness	0.23	0.06	0.29	*−*0.20	0.67
Neuroticism	0.80	0.22	0.00	0.40	1.21

Outcome variable is symptom load caused by SV experience. ^1^ Predictor variable is mean-centered; Lower BCa 95% CI and Upper BCa 95% CI: lower and upper limits of 95% bias-corrected and accelerated bootstrapped confidence interval of unstandardized regression coefficient B based on 1000 bootstrap samples.

**Table 8 healthcare-11-02501-t008:** Unstandardized indirect effects of length of professional experience as a pediatrician in years on symptom load caused by SV experience via Big Five personality traits.

	Unstandardized Effect	BootLLCI	BootULCI
Total	−0.006	−0.18	0.005
Openness	0.004	0	0.01
Conscientiousness	0	−0.003	0.004
Extraversion	0.001	−0.004	0.006
Agreeableness	−0.001	−0.005	0.002
Neuroticism	−0.01	−0.02	−0.003

BootLLCI, BootULCI: lower and upper limits of 95% confidence interval based on 5000 deviation-correction bootstrap samples.

**Table 9 healthcare-11-02501-t009:** Standardized indirect effects of length of professional experience as a pediatrician in years on symptom load caused by SV experience via Big Five personality traits.

	Standardized Effect	BootLLCI	BootULCI
Total	−0.02	−0.06	0.02
Openness	0.01	−0.001	0.03
Conscientiousness	0.002	−0.01	0.01
Extraversion	0.003	−0.004	0.02
Agreeableness	−0.002	−0.01	0.006
Neuroticism	−0.04	−0.07	−0.01

BootLLCI, BootULCI: lower and upper limits of 95% confidence interval based on 5,000 deviation-correction bootstrap samples.

## Data Availability

The data presented in this study are available on request from the corresponding author.

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
