# Peer review of "Second Victims among Austrian Pediatricians (SeViD-A1 Study)"

_healthcare, 2023, doi:10.3390/healthcare11182501_

Round 1

Reviewer 1 Report

Dear Authors,

thank you very much for your submission on this topic.

Your paper deals with an important topic and, as listed, is a good and important addition to the already published SeViD studies, with a focus on Austria.

The methods and results are well and scientifically prepared and presented to the reader in a comprehensible way.

Only the supplementary questions mentioned in line 70 to 72 ( Covid reference and fear for legal consequences). Find only rudimentary and somewhat misleading explanations in the following text.

Here the connection between recommended legal advice (line 257) after an event, with nevertheless only small fear of legal consequences (line 182), is not completely evident also after repeated reading. Likewise, the differentiation between classic support from a colleague and supportive legal advice and its added value could be explained more clearly and thus more comprehensibly.

Also, it would be beneficial for the general topic if you elaborate more clearly on the listed COVID reference in line 157 and emphasize that the pandemic was only an accelerant of the conditions and not, as still often emphasized, the trigger. This is indeed quite derivable from the events of your supplementary question. 

Thank you very much for your contribution and the possibility to judge it.

Author Response

Dear Authors,

thank you very much for your submission on this topic.

Your paper deals with an important topic and, as listed, is a good and important addition to the already published SeViD studies, with a focus on Austria.

The methods and results are well and scientifically prepared and presented to the reader in a comprehensible way.

Only the supplementary questions mentioned in line 70 to 72 ( Covid reference and fear for legal consequences). Find only rudimentary and somewhat misleading explanations in the following text.

You are right. We therefore added a paragraph in the discussion on COVID and added suggestions for legal advice. We inserted the paragraphs below for you to read.

Here the connection between recommended legal advice (line 257) after an event, with nevertheless only small fear of legal consequences (line 182), is not completely evident also after repeated reading. Likewise, the differentiation between classic support from a colleague and supportive legal advice and its added value could be explained more clearly and thus more comprehensibly.

Thanks for your suggestion. We changed the paragraph and added some more information. It now reads:

The rating of possible support measures showed that the most helpful support measure is the possibility to access legal consultation after a severe event. This contrasts to 65% of the participants stating that fear of legal consequences was only weakly if at all pronounced for them after experiencing SVP. Perhaps, in general, the fear of legal consequences is high but did not apply to the situations that caused the participants to become SVs. Still, this should be kept in mind when preparing a support program for Austrian pediatricians and access to legal advice should be offered after a severe adverse event complementary to systematic peer support, as suggested in Scott’s model [26].

Also, it would be beneficial for the general topic if you elaborate more clearly on the listed COVID reference in line 157 and emphasize that the pandemic was only an accelerant of the conditions and not, as still often emphasized, the trigger. This is indeed quite derivable from the events of your supplementary question. 

Thank you very much for this suggestion. We added a paragraph about the finding that COVID only played a role in 8% of the key events leading to SVP. The added part in the discussion now reads:

The added question if the key event leading to SVP was related to the COVID-pandemic showed that the pandemic did not trigger the majority of SVPs. Of course, the pandemic had a negative impact on health care workers mental health, leading to post-traumatic stress disorder or depression [24]. But in the case of SVP it seems to only play a minor role. The previous SeViD-studies have come to the same result: SeViD-I was conducted before the pandemic and the prevalence is similar to SeViD-II and -III which were conducted during the pandemic [14–16]. Therefore, the pandemic might be seen as an accelerator but not the main trigger of SVP.

Thank you very much for your contribution and the possibility to judge it.

Thank you very much for your valuable comments. We hope to have addressed everything as suggested.

Reviewer 2 Report

General comments

This is an interesting and important topic and the authors have presented data from an important group of medical practitioners. The sample is an opportunity sample of medical professionals in Austria using a list of Austrian Association for Pediatric and Youth Medicine (ÖGKJ), representing approximately 2, 100 members and the study achieved just less than a 20% response rate.

The introduction and rationale for the study could benefit from more evidence citations to support the theoretical models implied by the mediation analysis and the causal links between personality and experiences of the so-called ‘Second Victim Phenomenon’ (SVP).

The Measures described could also benefit from more detail as outlined in my comments below. There are a lot of results and I think some efficiencies in presentation could be achieved here.

The main thrust and causal language used in the paper should also be revised using more appropriate correlational and speculative statements about causality given the correlational nature of the data.

Specific comments

Abstract
1. Line 22 ‘1.000 samples’ should be 1000 or 1, 000.

2.     Line 75 states 2.100 members. Should be 2,100.

3.     Line 23-24
It states that ‘
High neuroticism and extraversion values as well as working in outpatient care were risk factors for SVP’. This implies that High N an E may predict SVP status but the relationship is merely correlational rather causally tested which means that those who experienced SV scored higher in N and E than those with no SV experience.

4.     Line 36. Full stop missing.

5.     Line 40 – remove the word Susan 

6.     Line 53 – remove ‘about SVP’ from sentence and add a space after the full-stop.

7.     The aim is to assess prevalence of SVP

8.     Line 59 remove the word ‘secondarily’ and replace with ‘secondly’.

9.     The survey used adaptive questioning: only those who have identified themselves as SVs were given the questions referring to the events leading to SVP and their symptoms. This may be a potential weakness in the study analysis as the symptoms of SVs cannot be compared with non-SVs.

Materials & Methods

10. Why was SVP status not dummy coded as 0 or 1 or consistency with in-patient status?

11.  Lines 92-95

It states that sums scores were used to calculate symptom load and that the items were scored 1, 0.5 and 0 and that ‘Don’t know’ was also scored as 0. Surely ‘Don’t know’ would be more accurately represented as ‘missing’  rather than given a zero symptom load? It would be useful to present how many indicated ‘Don’t know’ on each of the 13 symptom items (I suspect not many which is worth stating) and whether recoding the ‘Don’t knows’ as ‘missing’ changes the results in any meaningful way.

12.  In addition, the description of scoring should be revised to make it clear how may items were involved in the summed scores calculations. More details on the reactions measure (14 items and 4 domains) should be provided despite the authors’ reference to a previous German language paper detailing the measure. I am also left wondering why item scores were summed if there are 4 domains as described in a previous paper by Strametz et al., (2021) doi:10.1007/s40664-020-00400-y.

13.  Line 99 – changes ‘points’ to ‘point’.

14.  Time from critical event. A score of 6 should represent the longest time period but this may not be the case if the SV experiences occurred within a year of taking part in the study. Does re-scoring this 6 change the results.

Statistical analysis

15.  Two regression analyses are reported (a logistic regression with SV status on 9 predictors and a MLR with symptom load on the same 9 predictors). 

16.  Give a rationale / evidence base in the literature review for inclusion of each predictor in the models or state which are merely used as control variables.

17.  The main conceptual issue is that in the authors have specified a statistical model in which SV status is an outcome variable dependent upon various demographic and personality characteristics. This way of modelling the relationships implies that personality characteristics may uniquely predict SV status (controlling for the other demographic covariates (age, gender, professional experience, workplace). This further implies that personality characteristics (in this case, higher E and N scores) are risk factors for making medical mistakes but it may equally be the case that those who have made medical mistakes subsequently experience higher levels of anxiety and regret and this may partly be expressed in higher N scores. The correlational nature of the data should be acknowledged.

18.  A similar conceptual and statistical model is presented using symptom load as the outcome variable (using only those identified as SVs (n=365, I presume, although this is not explicitly stated in the multiple regression results summarised in table 7).  This table should include standardised coefficients as these give a comparative indication of effect sizes.

19.  I recommending changing the term ‘Independent Variable’ in tables 6 and 7 to ‘Predictor Variable’ and the term ‘Dependent Variable’ to ‘Outcome Variable’. These terms should also be replaced throughout the paper.

20.  Line 112

Be more explicit about which predictors were centered in the main text.

 21.  Lines 114-115

There is a need to state what tolerance / VIF values were used (with references) to infer collinearity.

 22.  Lines 116-118 states “we centered (weighted by the mean values) the predictor variables prior to conducting regression analyses if multicollinearity between the predictors was present”. This is not clear and needs to be re-phrased. The term ‘mean-centered’ is more efficient.

 23.  Line 119 cite the previous publication mentioned.

 24.  Line 125 ‘interval’ should be plural.

 25.  Line 126 change’5000 samples’ to ‘5000 bootstrap samples’.

 26.  Line 135 State the response rate here as well.

 27.  Line 139 , no need to state both male and female percentages.

 28.  Line 143 “working in in…” . Rephrase to avoid repetition.

 29.  Line 150 change ‘as shown I figure 1’ to ‘as shown in figure 1’.

 30.  Lines 151-2
It states that “…only 11% of the participants have stated to not have experienced this kind of traumatization”. This is poorly phrased. Rephrase and remove the word ‘only’.

31.  Figure 2 is redundant and could be removed and the figures added to table 1 for efficiency.

32.  Tables 2 and 3 could be combined for efficiency.

33.  Table 3 label is grammatically poor.

34.  Table 4 Figures should be refined to separate out the ‘I don’t know’ category from the ‘Not at all’ category. Bottom line missing from table 4.

35.  Lines 179-183

36.  It is not clear whether the percentages reported in relation to the ‘desire to be supported by others’, and ‘the desire to process the event for better understanding’ or ‘fear of legal consequences’ are part of table 5.

37.  Table 5 reports on means rather than percentages. There is a need to clarify the ‘abstentions’ column as this is not explained. This number is identical in all items in table 5 suggesting a sub-group of respondents who did not respond collectively to these items? More clarification is required here.

38.  Lines 191-92. ORs and their 95% CIs are sufficient in text. Remove B coefficients and 95% Cis from this text.

39.  Line 190 change text ‘proved to be a risk factors’ to ‘ were significant risk factors’.

40.  Line 196 change ‘referent’ to ‘reference’.

41.  Line 194 remove extra space tab.

42.  Line 201 ‘SVexperience change to ‘SV experience’.

43.  Mediation analysis
Given that professional experience (in years) was not significant in the multiple regression then a rationale for using it as a predictor in the mediation analysis should be provided. Perhaps the zero-order correlation between professional experience and symptom load was statistically significant?

44.  Change all reference to ‘independent’ and ‘dependent’ variables to ‘predictor’ and ‘outcome’ variables respectively. Clarify if b coefficients are standardised or unstandardised and report standardised coefficients for consistency with the previous multiple regression analysis.

45.  Table 8 likewise should report standardised effects for ease of comparison of effect sizes.

46.  It is not clear what the total and direct columns of table 8 add to the results of the parallel mediation analysis. Readers will be most interested to know
a. Which total indirect effects are significant,
b. The total indirect effect of professional experience (i.e. the sum of each total indirect effect for each mediator),
c. The total direct effect of professional experience on symptom load,
d. The total overall effect of professional experience on symptom load (i.e. total indirect plus total direct).

47.  Line 218 change ‘insignificant’ to ‘non-significant’.

48.  Line 214 spelling of ‘independent’.

49.  A table of Pearson zero-order correlations of all personality scales, professional experience and symptom scores would a be useful additional table.

Discussion

50.  I would suggest that prevalence is not the main aim of this study given the non-random nature of the sample and the response rate of less than 20% and the finding that 73.7% of participants had never heard of the term ‘Second Victim’. The emphasis of the discussion needs to be modified to reflect this.

51.  Line 258 and 263 change ‘didn’t’ to ’did not’ and can’t to ‘cannot’.

52.  Given the correlational nature of the data, I am sceptical of the conclusion that ‘that neuroticism seemed to be a risk factor for becoming a SV…’ and the subsequent causal statement that ‘neuroticism and openness seemed to increase the symptom load.’ In general, authors should refrain from using causal language (e.g. Line 293 ‘Multiple linear regression also showed that workplace has a significant influence on symptom load.’

53.  This are merely assumptions of the specified statistical models but these models are not compared to other competing models.

54.  The interpretation of the finding that greater ‘openness to experience’ correlated with higher symptom load could be explained by the possibility that those who are more open to experience are perhaps more curious and willing to explore different avenues of treatment or intervention which may have higher risk for patients.

55.  Authors should also attempt to explain the apparent contradiction in the findings that higher E scores were associated with a greater likelihood of experiencing SV but lower symptom load scores.

56.  Line 282 states that  neuroticism mediates the influence of work experience on symptom load.’ This should be further explained. Does this mean that greater work experience is associated with both lower symptom load and lower N? A table of correlations of all personality scales, professional experience and symptom scores would a be useful additional table in the results to help elucidate the overall negative ‘mediated effect’ of professional experience on symptom load.

57.  Line 284 has an unfinished sentence.

Some minor improvements required as suggested in my specific comments 

Author Response

General comments

This is an interesting and important topic and the authors have presented data from an important group of medical practitioners.

We appreciate the time and effort that you have dedicated to reviewing our manuscript. We have read your suggestions carefully and amended the text accordingly. We believe that the quality of the manuscript has been improved in this way and hope that we could rule out any concerns on our diligence.

The sample is an opportunity sample of medical professionals in Austria using a list of Austrian Association for Pediatric and Youth Medicine (ÖGKJ), representing approximately 2, 100 members and the study achieved just less than a 20% response rate.

We have added this as a limitation of the study. It now reads:

“Of the approximately 2.100 contacted members of ÖGKJ, only 414 participated in this study. Fosnacht et al. showed that the response rate of 20% could provide reliable estimates in medium sample sizes (n=300-500) applying the convenience sampling method [55].”

The introduction and rationale for the study could benefit from more evidence citations to support the theoretical models implied by the mediation analysis and the causal links between personality and experiences of the so-called ‘Second Victim Phenomenon’ (SVP).

We added the rationale for the regression as well as mediation analysis in the introduction section. In addition, we understand that certain authors have their reservations regarding the usage of the term and concept of Second Victim and that they might considered it as “putting old wine in a new bottle”. However, this term is widely used, the consent definition of the phenomenon has been introduced and the measurements of SVP have been validated in many languages[4]. We strongly believe that the trend toward establishing the term and concept in the science is imminent similar to burnout that has been doubted as a novel construct after its introduction[4,5]. We also believe that even though many health care professionals are not familiar with this term (as shown in our study) at the moment, the common usage of the term would increase as the awareness of the concept increases among this population. We discuss this point in the discussion section.

The Measures described could also benefit from more detail as outlined in my comments below.

Thank you for your comment. We described the measures in the method section in more detail.

There are a lot of results and I think some efficiencies in presentation could be achieved here.

We adapted the presentation of our results to provide additional efficiencies.

The main thrust and causal language used in the paper should also be revised using more appropriate correlational and speculative statements about causality given the correlational nature of data.

We also adapted the language by weakening the statements regarding causality throughout the manuscript. However, using regression analysis we always at least theoretically indicate the direction of the correlation between the regressor and regressand. We explained the suggested interpretation of the causal association in more detail throughout the revised manuscript.

Specific comments

Abstract
1. Line 22 ‘1.000 samples’ should be 1000 or 1, 000.

We changed it as you suggested to 1,000.

  1. Line 75 states 2.100 members. Should be 2,100.

Also changed to 2,100.

  1. Line 23-24
    It states that ‘High neuroticism and extraversion values as well as working in outpatient care were risk factors for SVP’. This implies that High N an E may predict SVP status but the relationship is merely correlational rather causally tested which means that those who experienced SV scored higher in N and E than those with no SV experience.

You are right. Our study design does not allow causal predictions. We therefore changed the line to:

“High neuroticism and extraversion values as well as working in outpatient care positively correlated to having experienced SVP.”

  1. Line 36. Full stop missing.

Changed, thank you.

  1. Line 40 – remove the word Susan

We removed it as suggested.

  1. Line 53 – remove ‘about SVP’ from sentence and add a space after the full-stop.

Changed both, thank you very much for your precise review.

  1. The aim is to assess prevalence of SVP

Changed it. The sentence now reads:

“To summarize, the study’s primary aim is to assess prevalence, symptom load and preferred support strategies of SVP and secondly to identify demographic, workplace-related and personality trait factors that significantly correlate to the likelihood of experiencing SVP and the amount and severity of possible symptoms caused by SVP amongst Austrian pediatricians . [11, 12]. Lastly, our aim is to test the mediational model that postulates that length of professional experience can indirectly reduce symptom load caused by the SV experience through Big-five personality traits”.

  1. Line 59 remove the word ‘secondarily’ and replace with ‘secondly’.

Changed it, thank you very much.

  1. The survey used adaptive questioning: only those who have identified themselves as SVs were given the questions referring to the events leading to SVP and their symptoms. This may be a potential weakness in the study analysis as the symptoms of SVs cannot be compared with non-SVs.

Thank you for this remark. We acknowledge this as one of the study’s limitations. The paragraph reads:

“Additionally, by using adaptive questioning the study does not allow comparison between those participants who have experienced SVP and those who have not.”

Materials & Methods

  1. Why was SVP status not dummy coded as 0 or 1 or consistency with in-patient status?

 We corrected this. SVP is dummy coded as 0 `no SVP experienced` and 1 `SVP at least once experienced` and is consistent to the item referring to in-patient status.

  1. Lines 92-95

It states that sums scores were used to calculate symptom load and that the items were scored 1, 0.5 and 0 and that ‘Don’t know’ was also scored as 0. Surely ‘Don’t know’ would be more accurately represented as ‘missing’ rather than given a zero symptom load? It would be useful to present how many indicated ‘Don’t know’ on each of the 13 symptom items (I suspect not many which is worth stating) and whether recoding the ‘Don’t knows’ as ‘missing’ changes the results in any meaningful way.

  In accordance with our previous studies, we scored ‘don’t know’ response option together with ‘not at all’. But we repeated our analyses treating ‘I don’t know‘ as listwise missing values. We report on them in the supplementary material.

  1. In addition, the description of scoring should be revised to make it clear how may items were involved in the summed scores calculations. More details on the reactions measure (14 items and 4 domains) should be provided despite the authors’ reference to a previous German language paper detailing the measure. I am also left wondering why item scores were summed if there are 4 domains as described in a previous paper by Strametz et al., (2021) doi:10.1007/s40664-020-00400-y .

 We explained this issue in the method section.

  1. Line 99 – changes ‘points’ to ‘point’.

We deleted this part of the sentence, see below.

  1. Time from critical event. A score of 6 should represent the longest time period but this may not be the case if the SV experiences occurred within a year of taking part in the study. Does re-scoring this 6 change the results.

You are absolutely correct. As we performed no inferential statistics using this item, we changed the text from “using a 6-points Likert scale” to “recorded as follows”, emphasizing that the points were not scored in an order of rank.

Statistical analysis

  1. Two regression analyses are reported (a logistic regression with SV status on 9 predictors and a MLR with symptom load on the same 9 predictors).

Yes, we used the same predictors in both binary-logistic and multiple linear regression. We explained that in the manuscript.

  1. Give a rationale / evidence base in the literature review for inclusion of each predictor in the models or state which are merely used as control variables.

We included no control variables as we were interested in the variance explanation and predictive value of all included “predictors” (predictor as well as independent variable is also not very adequate term in cross-sectional design, but they are widely used as synonyms, see section 19). We provided rationale for every included variable in the introduction section.

  1. The main conceptual issue is that in the authors have specified a statistical model in which SV status is an outcome variable dependent upon various demographic and personality characteristics. This way of modelling the relationships implies that personality characteristics may uniquely predict SV status (controlling for the other demographic covariates (age, gender, professional experience, workplace). This further implies that personality characteristics (in this case, higher E and N scores) are risk factors for making medical mistakes but it may equally be the case that those who have made medical mistakes subsequently experience higher levels of anxiety and regret and this may partly be expressed in higher N scores. The correlational nature of the data should be acknowledged.

In SeViD I[6] survey only demographic, and work-place related variables were included in the model. In the SeViD II[7] and III[8] we also included personality traits. Our intention in the current as well in the SeViD III Study[8] (which is available in English language see the listed references below) was to test if the personality traits can significantly contribute to the likelihood prediction of SVP or the variance explanation of the symptom load beyond the variance explained by demographic and work-place related variables since the latter had already been tested in the SeViD I study. We believe that our causal assumption (regression analyses always imply causality in the statistical model, even in the cross-sectional design) is theory-driven since personality traits remain relatively stable over the lifespan among the adult population[9,10]. Moreover, neuroticism was found to remain relatively stable even after a traumatic event[11]. In conclusion, we believe that our interpretation of neuroticism effecting the likelihood of SV experience and the symptom load is more plausible than vice versa. We refer to this problematic in the discussion section. In addition, although we have explained that we cannot demonstrate causality in our original submitted paper in the limitations section, we acknowledged the correlational nature of the data throughout the manuscript.

  1. A similar conceptual and statistical model is presented using symptom load as the outcome variable (using only those identified as SVs (n=365, I presume, although this is not explicitly stated in the multiple regression results summarised in table 7). This table should include standardised coefficients as these give a comparative indication of effect sizes.

We used n=313 participants, we added this information in the manuscript. We elected to report only on unstandardized regression coefficients in our original submission because the bootstrapping procedure using SPSS reveal only the significance and BCa 95%CI of the unstandardized regression coefficients. The standardized coefficients and their BCa 95%CI are not provided in the output for the bootstrapping analysis. However, we accepted your suggestion, as the non-bootstrapped and bootstrapped standardized coefficients did not differ in our analysis (probably due to the large sample size and according to the central limit theorem) and added a column with standardized regression coefficients in the table 7.  

  1. I recommending changing the term ‘Independent Variable’ in tables 6 and 7 to ‘Predictor Variable’ and the term ‘Dependent Variable’ to ‘Outcome Variable’. These terms should also be replaced throughout the paper.

 This point is debatable because neither (in)dependent variable nor predictor/outcome terminology is theoretically accurate in terms of cross-sectional study design. The problem is that theoretically correct terms explanans and explanandum as well as regressand and regressor are barely used in the publications. Nevertheless, we accepted your proposal after careful consideration because we used two different terminologies, which can suggest incoherency. We use predictor/outcome terms throughout the revised manuscript.

  1. Line 112

Be more explicit about which predictors were centered in the main text.

We reported on centered predictors more explicitly.

  1. Lines 114-115

There is a need to state what tolerance / VIF values were used (with references) to infer collinearity.

We added the reference.

  1. Lines 116-118 states “we centered (weighted by the mean values) the predictor variables prior to conducting regression analyses if multicollinearity between the predictors was present”. This is not clear and needs to be re-phrased. The term ‘mean-centered’ is more efficient.

We corrected this.

  1. Line 119 cite the previous publication mentioned.

Thank you, we added the citation.

  1. Line 125 ‘interval’ should be plural.

We corrected this.

  1. Line 126 change’5000 samples’ to ‘5000 bootstrap samples’.

We changed this.

  1. Line 135 State the response rate here as well.

We reported the response rate.

  1. Line 139 , no need to state both male and female percentages.

That was indeed necessary since we had “diverse(non-binary) gender” response alternative in the questionnaire, we report on that in the revised manuscript.

  1. Line 143 “working in in…” . Rephrase to avoid repetition.

We rephrased this.

  1. Line 150 change ‘as shown I figure 1’ to ‘as shown in figure 1’.

We changed this.

  1. Lines 151-2
    It states that “…only 11% of the participants have stated to not have experienced this kind of traumatization”. This is poorly phrased. Rephrase and remove the word ‘only’.

 We rephrased the sentence.

  1. Figure 2 is redundant and could be removed and the figures added to table 1 for efficiency.

We appreciate your suggestion, but we elected not to remove the figure 2. In other previously published SeViD studies a pie chart showing the second victim prevalence was an integral element of the descriptive statistics results section. We believe that the readers can get a better impression about the topic of the survey. Therefore, we would like to keep the figure in the current study, too.

  1. Tables 2 and 3 could be combined for efficiency.

We combined the tables 2 and 3.

  1. Table 3 label is grammatically poor.

We changed the label.

  1. Table 4 Figures should be refined to separate out the ‘I don’t know’ category from the ‘Not at all’ category. Bottom line missing from table 4.

We added a separate ‘I don’t know’ category and the bottom line.

  1. Lines 179-183
  2. It is not clear whether the percentages reported in relation to the ‘desire to be supported by others’, and ‘the desire to process the event for better understanding’ or ‘fear of legal consequences’ are part of table 5.

No, they are not part of the table 5. Only symptoms included in the symptom load sum score are presented in the table 5. ‘Desire to be supported by others’, and ‘the desire to process the event for better understanding’ were not included in computing the overall sum score, because we treated them as a grey area between symptoms and support desires according to the SeViD III study.Fear of legal consequences’ was firstly introduced as a symptom in this study and was not a part of previous symptom load sum score.  However, we reported on them in the results section. We added a footnote in the Table 3 stating that only symptoms included in the symptom load score were listed in the table 4 (table 3 in the revised manuscript),

  1. Table 5 reports on means rather than percentages. There is a need to clarify the ‘abstentions’ column as this is not explained. This number is identical in all items in table 5 suggesting a sub-group of respondents who did not respond collectively to these items? More clarification is required here.

The missing values were all dropouts, who quit the participations on the study before the support block. We reported that and we revised the table 5.

  1. Lines 191-92. ORs and their 95% CIs are sufficient in text. Remove B coefficients and 95% Cis from this text.

Thank you for your comment. However, we decided not to remove B coefficients and 95% Cis as this parameter is very important since the bootstrapped estimates refer only to B coefficients and not to ORs. The assumptions for logistic regressions do not refer to neither normally distribution nor variance homogeneity (homoscedasticity).  Nevertheless, certain assumptions must be met to perform logistic regressions. They include apart from independency of observations and absence of multicollinearity, the linearity in the logit for continuous variables as well as absence of strongly influential outliers[12]. Validation of these assumptions can lead to inaccurate parameters estimation and standard errors. Another problem, that can occur using logistic regression model refer to data overfitting. In logistic regressions with many predictors, like in our case, the coefficients tend to be different from zero. Thus, the obtained effect sizes can be artificially increased. However, using the bootstrapping approach we can achieve robust estimates for parameter coefficients, confidence intervals and standard errors and solve the problem of overfitting[13].

  1. Line 190 change text ‘proved to be a risk factors’ to ‘ were significant risk factors’.

Changed, thank you.

  1. Line 196 change ‘referent’ to ‘reference’.

We corrected this, thank you.

  1. Line 194 remove extra space tab.

We removed the extra space tab.

  1. Line 201 ‘SVexperience change to ‘SV experience’.

Changed, thank you.

  1. Mediation analysis
    Given that professional experience (in years) was not significant in the multiple regression then a rationale for using it as a predictor in the mediation analysis should be provided. Perhaps the zero-order correlation between professional experience and symptom load was statistically significant?

We provided the rationale for the mediation analysis in the introduction section.

  1. Change all reference to ‘independent’ and ‘dependent’ variables to ‘predictor’ and ‘outcome’ variables respectively. Clarify if b coefficients are standardised or unstandardised and report standardised coefficients for consistency with the previous multiple regression analysis.

We changed the terminology and we clarified the b coefficients,

  1. Table 8 likewise should report standardised effects for ease of comparison of effect sizes.

We report on the standardized indirect effects because they are the only presented in the SPSS Process Output.

  1. It is not clear what the total and direct columns of table 8 add to the results of the parallel mediation analysis. Readers will be most interested to know
    a. Which total indirect effects are significant,
    b. The total indirect effect of professional experience (i.e. the sum of each total indirect effect for each mediator),
    c. The total direct effect of professional experience on symptom load,
    d. The total overall effect of professional experience on symptom load (i.e. total indirect plus total direct).

Table 8 and shows the effects of professional experience on symptom load via each of the Big-five. We accepted your opinion and added information about standardized and unstandardized indirect effects in the table 8 and 9. Total direct as well as overall effect are presented in the text.

  1. Line 218 change ‘insignificant’ to ‘non-significant’.

We corrected this.

  1. Line 214 spelling of ‘independent’.

Changed, thank you.

  1. A table of Pearson zero-order correlations of all personality scales, professional experience and symptom scores would a be useful additional table.

We added the table.

Discussion

  1. I would suggest that prevalence is not the main aim of this study given the non-random nature of the sample and the response rate of less than 20% and the finding that 73.7% of participants had never heard of the term ‘Second Victim’. The emphasis of the discussion needs to be modified to reflect this.

We discuss this issue in the discussion section. We relativized the usage of the term prevalence,

  1. Line 258 and 263 change ‘didn’t’ to ’did not’ and can’t to ‘cannot’.

We changed it accordingly.

  1. Given the correlational nature of the data, I am sceptical of the conclusion that ‘that neuroticism seemed to be a risk factor for becoming a SV…’ and the subsequent causal statement that ‘neuroticism and openness seemed to increase the symptom load.’ In general, authors should refrain from using causal language (e.g. Line 293 ‘Multiple linear regression also showed that workplace has a significant influence on symptom load.’)

Thank you for your important input. Of course, we cannot make causal connections in this type of study. We therefore changed the wordings accordingly. The paragraph now reads:

The examination of the Big Five personality traits in regards to SVP showed that neuroticism positively correlates with becoming a SV which was also discovered in SeViD-II and -III: Participants with higher neuroticism levels were more likely to be a SV than those with lower levels. A new finding of this study is, that extraversion, though shown to positively correlate with becoming a SV, seems to negatively correlate with the symptom load when being affected by SVP, while neuroticism and openness seemed to correlate with higher symptom load.

We also carefully revised the entire manuscript and changed causal language accordingly.

  1. This are merely assumptions of the specified statistical models but these models are not compared to other competing models.

We discuss this issue in the discussion section.

  1. The interpretation of the finding that greater ‘openness to experience’ correlated with higher symptom load could be explained by the possibility that those who are more open to experience are perhaps more curious and willing to explore different avenues of treatment or intervention which may have higher risk for patients.

Thank you for this valuable input. We have not thought of this before, but of course, it makes sense. We therefore added the following paragraph:

A possible explanation might be that those with greater openness might be more likely to explore alternative treatments or interventions. If these interventions or treatments are not as established, they might bare higher risks to patients and therefore lead to more adverse events when using them. This could not only trigger SVP but also increase the symptom load since those with higher levels of openness might feel more guilty after choosing the riskier treatment or intervention for their patients.

  1. Authors should also attempt to explain the apparent contradiction in the findings that higher E scores were associated with a greater likelihood of experiencing SV but lower symptom load scores.

We provided a possible explanation for the finding.

  1. Line 282 states that ‘neuroticism mediates the influence of work experience on symptom load.’ This should be further explained. Does this mean that greater work experience is associated with both lower symptom load and lower N? A table of correlations of all personality scales, professional experience and symptom scores would a be useful additional table in the results to help elucidate the overall negative ‘mediated effect’ of professional experience on symptom load.

We added the table with Pearson’s correlations.

  1. Line 284 has an unfinished sentence.

Thank you very much for your observance. We finished the sentence. It now reads:

“Therefore, it is possible, that neurotic people are more likely to leave their jobs meaning that a selection effect might have taken place for those with higher work experience.”

References

  1. Management, O.o.; Budget. Questions and answers when designing surveys for information collections. 2006.
  2. Wu, M.-J.; Zhao, K.; Fils-Aime, F. Response rates of online surveys in published research: A meta-analysis. Computers in Human Behavior Reports 2022, 7, 100206, doi:https://doi.org/10.1016/j.chbr.2022.100206.
  3. Fosnacht, K.; Sarraf, S.; Howe, E.; Peck, L. How Important are High Response Rates for College Surveys? The Review of Higher Education 2017, 40, 245-265, doi:10.1353/rhe.2017.0003.
  4. Seys, D.; Panella, M.; Russotto, S.; Strametz, R.; Mira, J.; Wilder, A.; Godderis, L.; Vanhaecht, K. In search of an international multidimensional action plan for second victim support: a narrative review. BMC Health Services Research 2023, 23, doi:10.1186/s12913-023-09637-8.
  5. Maslach, C. Engagement research: Some thoughts from a burnout perspective. European Journal of Work and Organizational Psychology 2011, 20, 47-52, doi:10.1080/1359432X.2010.537034.
  6. Strametz, R.; Koch, P.; Vogelgesang, A.; Burbridge, A.; Rösner, H.; Abloescher, M.; Huf, W.; Ettl, B.; Raspe, M. Prevalence of second victims, risk factors and support strategies among young German physicians in internal medicine (SeViD-I survey). J Occup Med Toxicol 2021, 16, 11, doi:10.1186/s12995-021-00300-8.
  7. Strametz, R.; Fendel, J.C.; Koch, P.; Roesner, H.; Zilezinski, M.; Bushuven, S.; Raspe, M. Prevalence of Second Victims, Risk Factors, and Support Strategies among German Nurses (SeViD-II Survey). Int J Environ Res Public Health 2021, 18, doi:10.3390/ijerph182010594.
  8. Marung, H.; Strametz, R.; Roesner, H.; Reifferscheid, F.; Petzina, R.; Klemm, V.; Trifunovic-Koenig, M.; Bushuven, S. Second Victims among German Emergency Medical Services Physicians (SeViD-III-Study). Int J Environ Res Public Health 2023, 20, doi:10.3390/ijerph20054267.
  9. Rantanen, J.; Metsäpelto, R.L.; Feldt, T.; Pulkkinen, L.; Kokko, K. Long-term stability in the Big Five personality traits in adulthood. Scand J Psychol 2007, 48, 511-518, doi:10.1111/j.1467-9450.2007.00609.x.
  10. Hampson, S.E.; Goldberg, L.R. A first large cohort study of personality trait stability over the 40 years between elementary school and midlife. J Pers Soc Psychol 2006, 91, 763-779, doi:10.1037/0022-3514.91.4.763.
  11. Ogle, C.M.; Rubin, D.C.; Siegler, I.C. Changes in neuroticism following trauma exposure. J Pers 2014, 82, 93-102, doi:10.1111/jopy.12037.
  12. Schreiber-Gregory, D.N.; Jackson, H.M.; Bader, K. Logistic and Linear Regression Assumptions : Violation Recognition and Control. 2018.
  13. Fernandez-Felix, B.M.; García-Esquinas, E.; Muriel, A.; Royuela, A.; Zamora, J. Bootstrap internal validation command for predictive logistic regression models. The Stata Journal 2021, 21, 498-509, doi:10.1177/1536867x211025836.

Round 2

Reviewer 2 Report

General comments

This is a much improved manuscript and the authors have attempted to comment on all the points made in my initial review.

Abstract

Change to ‘with’ underlined.

“High neuroticism and extraversion values as well as working in outpatient care positively correlated with having experienced SVP.”

"To summarize, the study’s primary aim is to assess prevalence, symptom load and preferred support strategies of SVP and secondly to identify demographic, workplace-related and personality trait factors that significantly correlate with the likelihood of experiencing SVP and the amount and severity of possible symptoms caused by SVP amongst Austrian pediatricians . [11, 12]. Lastly, our aim is to test the mediational model that postulates that length of professional experience can indirectly reduce symptom load caused by the SV experience through Big-five personality traits.

Materials & Methods

The description of scale scoring has been extensively revised and a rationale is given for the coding of ‘don’t know’ as 0.

Mediation analysis

Table 8
The term ‘Total unstandardised indirect effect’ label in column 1 could be reduced to ‘Total’ as column 1 explains that it’s the ‘Unstandardized Effect’.

Why are the terms ‘Total’, ‘Extraversion’ and ‘Agreeableness’ in italics?

Discussion

Consider soften the term ‘neurotic people’ to ‘those scoring higher on neuroticism’. I am not convinced that a self-report personality measure is a diagnostic instrument.

Author Response

General comments

This is a much improved manuscript and the authors have attempted to comment on all the points made in my initial review.

Thank you very much. We appreciated your comments and believe the extended revision truly did improve the quality of our manuscript.

Abstract
Change to ‘with’ underlined.

“High neuroticism and extraversion values as well as working in outpatient care positively correlated with having experienced SVP.”

We changed it as you suggested.

"To summarize, the study’s primary aim is to assess prevalence, symptom load and preferred support strategies of SVP and secondly to identify demographic, workplace-related and personality trait factors that significantly correlate with the likelihood of experiencing SVP and the amount and severity of possible symptoms caused by SVP amongst Austrian pediatricians . [11, 12]. Lastly, our aim is to test the mediational model that postulates that length of professional experience can indirectly reduce symptom load caused by the SV experience through Big-five personality traits.”

Thank you for this suggestion, the sentence reads as you suggest, but we did not add “[11,12]” because the references do not fit the content of the paragraph – perhaps this was an editing error?

Materials & Methods

The description of scale scoring has been extensively revised and a rationale is given for the coding of ‘don’t know’ as 0.

Mediation analysis

Table 8
The term ‘Total unstandardised indirect effect’ label in column 1 could be reduced to ‘Total’ as column 1 explains that it’s the ‘Unstandardized Effect’.

We changed the label to “total”, as you suggested. Likewise, we changed the label in table 9 accordingly.

Why are the terms ‘Total’, ‘Extraversion’ and ‘Agreeableness’ in italics?

We changed this. Must have been an editing error – thank you.

Discussion

Consider soften the term ‘neurotic people’ to ‘those scoring higher on neuroticism’. I am not convinced that a self-report personality measure is a diagnostic instrument.

We softened this as you suggested in that paragraph of the discussion.